# How do public services supply, livelihood capital, and livelihood strategies affect subjective poverty?

Yuanquan Lu[1,2], Li Chen [1]*, Yuan Meng[1]

1 School of Public Policy and Administration, Chongqing University, Chongqing, China, 2 School of Economics and Management, Chongqing Normal University, Chongqing, China

☯ These authors contributed equally to this work.

* leilachenli@163.com

**Data Availability Statement:** All data files are available from the Chinese General Social Survey database (http://cgss.ruc.edu.cn/index.htm).

**Funding:** The author(s) received no specific funding for this work.

## Abstract

Poverty is not only an economic problem but also a social problem, and there are certain limitations of objective poverty based on the population's income. It does not reflect the residents' true feelings regarding education opportunities, pension and medical security, and participation in decision-making. Researchers have studied it intensively in different objective dimensions of Chinese poverty, and little attention has been paid to subjective poverty. This study analyzes how public services supply, livelihood capital, and livelihood strategies affect subjective perceptions of poverty. The results show that public services supply, livelihood capital, and livelihood strategies significantly correlate with subjective poverty. Physical capital and social capital have the greatest effects on the occurrence of subjective poverty. The probability of subjective poverty decreases by 0.149 and 0.107 for each unit change in physical and social capital, respectively. What's more, public services supply, physical capital, financial capital, and human capital affect the subjective poverty of urban and rural residents at different significance levels. It means that the formation of subjective poverty results from the superposition of multiple factors.

## Introduction

Poverty governance is a critical challenge faced by numerous developing countries worldwide. The United Nations' Sustainable Development Goals have prioritized "No Poverty" as the first goal, aiming to "End poverty in all its forms everywhere" by 2030 [1]. China is also committed to this goal. Over the last four decades, China has successfully implemented the "Targeted Poverty Alleviation Strategy," a campaign that has yielded remarkable results in poverty governance and made significant contributions to global poverty reduction efforts. This strategy has effectively counteracted the adverse impact of economic growth decline and income inequality on poverty. As per current standards, an impressive 98.99 million rural individuals have been lifted out of poverty in China. Additionally, 832 poverty-stricken counties and 128,000 impoverished villages have been successfully uplifted. China has effectively tackled overall regional poverty, and this accomplishment has substantially improved people's capacity to work and

**Competing interests:** The authors have declared that no competing interests exist.

thrive. Nevertheless, even with the objective poverty issue addressed, subjective poverty has emerged as a prominent social problem in China's pursuit of constructing a comprehensive socialist modern state [2]. Therefore, it becomes crucial to devise follow-up plans and measures to sustain the achievements made in poverty elimination. In a similar context, Mahmood et al. discovered discrepancies between objective and subjective poverty in Pakistan, attributable to various influencing factors [3]. Understanding and addressing such nuances are pivotal for crafting effective poverty governance strategies.

Subjective poverty refers to individuals' self-perception of poverty, representing their personal assessment of living conditions rather than an external determination. The concept of subjective poverty arose as a rethinking of the objective poverty line in academic history, with Dutch scholars pioneering subjective poverty line measurement in the 1970s. It is important to note that subjective poverty and objective poverty are not equivalent. Koczan proposed a dual definition of poverty: one is objective economic poverty, where income is insufficient to cover daily expenses, and the other is psychological subjective poverty [4]. The subjective evaluation of poverty takes into account various factors such as property and income, access to education, pension and medical security, and participation in decision-making processes. In contrast to objective poverty, subjective poverty offers a more comprehensive assessment, considering an individual's overall attributes and values. This includes residents' perceptions of income distribution, the allocation of educational resources, and the effectiveness of social security systems. Subjective poverty also encompasses the overall evaluation of policy implementation. In China, the notion of equal rights in income distribution holds significant value.

The Chinese government has taken significant steps toward improving the welfare of its citizens. Their efforts have focused on promoting equal educational opportunities, providing comprehensive social security coverage, and enhancing the overall sense of acquisition, security, and happiness among residents. As pointed out by Koczan, addressing both residents' objective economic poverty and psychologically subjective poverty is crucial to minimizing resistance to reform and development [4]. Consequently, beyond solving the issue of objective poverty, eradicating subjective poverty has become a major political agenda in China after 2020. To achieve this, we utilized data from the Chinese General Social Survey in 2019 to study the influencing factors of subjective poverty. This research is instrumental in assisting the government in timely policy adjustments and facilitating social and economic reforms.

## Literature review

According to the existing research results, most of the literature discusses the impact of differences in household micro-features on subjective poverty from multiple aspects, such as population, economy, and policy. Scholars have found that demographic characteristics, including family consumption, household size, marital status, gender differences, educational background, and ancestry, are significantly related to subjective poverty [2, 3, 5]. Nándori [6] analyzes subjective poverty in Hungary and finds that Roma descent, entitlement to social support, and unemployment affect subjective poverty. Alem et al. [7] find that families with a history of poverty will continue to think they are in poverty even if their material consumption increases. Mahmood et al. [3] use Pakistan Panel Household Survey 2010 data, and they find that the determinants of subjective poverty are not limited to household consumption but include physical health, food security, sanitation facility, household size, household demographic structure, and agriculture land ownership.

The economic aspect mainly involves the relationship between subjective poverty and factors such as socioeconomic background, income level, household disposable income, and employment status [8]. It has been found that all of these factors significantly affect the

occurrence of subjective poverty directly or indirectly [1, 3]. Joyce and Ziliak [9] compare the causes of relative poverty in Britain and the United States and find that relative poverty in Britain is related to working households, while low-skilled is the main cause of relative poverty in the United States. Some scholars have also studied the relationship between policy and poverty. Li and Li [10] find that the anti-poverty relocation and settlement program affects subjective well-being by affecting the probability that people will be exposed to risks due to policy. Agasisti et al. [11] collected relevant data from European countries between 2006 and 2015 and found a significant correlation between education and poverty. Zheng and Wang [12] discovered that social pension insurance can effectively reduce poverty among 95% of rural elderly individuals.

In summary, the papers and opinions discussed in this context offer valuable insights into understanding subjective poverty. However, certain research gaps still exist. Many studies primarily focus on the impact of individual factors on subjective poverty, overlooking the interconnectedness between various factors and their combined effect on subjective poverty. Additionally, there is a dearth of research exploring the influence of livelihood capital on subjective poverty. As subjective poverty is more comprehensive than objective poverty, it eventually encompasses multidimensional aspects of poverty. To contribute to the theoretical research in this area, our study aims to analyze subjective perceptions of poverty in China. Specifically, we employ logistic regression and the "self-rating scale" method, using public services supply, subsistence capital, and livelihood strategies as the measurement dimensions to investigate the impact of subjective poverty in this region. The results of this research could assist the government in making timely policy adjustments and promoting social and economic reforms. Moreover, it serves as a valuable addition to empirical research on subjective poverty, offering a more comprehensive understanding of the complexities involved in assessing and addressing poverty from a subjective perspective.

## Theoretical basis and research hypothesis

The concept of "sustainable livelihoods" has its origins in the expansion and reflections of anti-poverty research by eminent scholars such as Amartya Sen [13]. Towards the end of the 20th century, Chambers and Conway introduced the term "livelihood" for the first time, defining it as a comprehensive concept encompassing an individual's ability to earn a living, the assets they possess, the external support available to them, and the actions they undertake to generate income for long-term improvements in their production and living conditions [14]. This concept highlights that at its core, livelihood involves "the action and process of converting livelihood capital through the exchange of various forms of capital" [3]. Capital conversion aims to foster sustainable and continuous livelihood expansion. Having abundant livelihood capital can lead to improved personal production and living standards, bolstering one's ability to cope with risks and breaking the cycle of intergenerational poverty transmission. Furthermore, different types of livelihood capital can be transformed into each other, giving rise to various welfare states depending on how they are utilized and exchanged. This interplay of livelihood capital forms the basis for creating resilient and thriving livelihoods.

In the report "*Our Common Future*", Brundtland first introduced the concept of sustainable livelihoods during the 1987 United Nations Conference on Environment and Development. In the 1990s, this concept underwent further development and clarification by Western scholars, leading to the emergence of the sustainable livelihoods approach. International agencies such as the United Nations Development Programme, the Department for International Development, and the Food and Agriculture Organization of the United Nations adopted this approach as a method to comprehensively understand livelihoods, assess the influence of

livelihood strategies on poverty, and identify appropriate interventions to address poverty-related challenges. Among the various frameworks proposed for sustainable livelihoods, the one put forth by the Department for International Development has gained widespread acceptance and is most commonly used in the Sustainable Livelihoods Guide [15]. We will utilize this framework to explore the interactions between four types of public services supply, livelihood capital, and livelihood strategies, which include physical capital, human capital, social capital, and financial capital. By employing this theory, we aim to shed light on how individual livelihood systems impact subjective poverty, providing valuable insights for addressing poverty-related issues and promoting sustainable development[①].

## Public services supply and subjective poverty

Appropriate public service supply can play a crucial role in addressing the issue of subjective poverty to a certain extent. In the 1970s and 1980s, Western social sciences underwent a "rediscovery" of the significance and role of institutional analysis in explaining practical problems, leading to the development of the new institutionalist analysis paradigm. At the heart of new institutionalism lies the concept of a "system," which encompasses the various rules that govern human social life, including both formal and informal rules, as well as operational mechanisms. In the context of poverty governance, the system serves as a transformative mechanism through which the public sector designs and integrates various resources to achieve rational poverty alleviation. Public services supply is an integral part of this institutional arrangement and policy practice. It is directly linked to the redistribution process of public resources. However, when the public services supply is insufficient and unbalanced, it results in a relative lack of public welfare rights. This, in turn, becomes a significant obstacle to promoting the flow of resources, enhancing development capabilities, and ultimately eradicating poverty. Addressing the challenges in public services supply and ensuring a more equitable distribution of resources is essential to effectively combat subjective poverty and facilitate overall socioeconomic development. By strengthening public services and making them more accessible to all segments of society, governments can play a critical role in reducing poverty and enhancing the well-being of their citizens. Basic public services play a crucial role in empowering the poor to overcome internal and external challenges. One of the significant ways in which these services help is by assisting the poor in finding employment opportunities, thereby expanding their capabilities and possibilities. By providing essential support, public services can greatly encourage and inspire the impoverished population, enabling them to break free from the shackles of poverty.

Hypothesis 1: The more perfect the government's public services supply is, the better able it is to reduce the occurrence of subjective poverty.

## Livelihood capital and subjective poverty

The main manifestation of subjective poverty is the lack of livelihood capital, which encompasses a full or partial shortage of physical capital, financial capital, human capital, and social capital [16]. Currently, poverty measurement primarily focuses on the amount of physical and financial capital, as they form the foundation of livelihood capital and influence the acquisition of other forms of capital. The scarcity of human capital is evident at both regional and individual levels. The rapid pace of industrialization and urbanization has led to the migration of many skilled laborers to big cities, leaving rural areas or underdeveloped cities with a limited number of low-skilled laborers. Combined with various factors like environmental conditions and resource availability, this situation hinders fundamental internal motivation for economic development and makes certain groups susceptible to subjective poverty.

Furthermore, social capital also plays a role in perpetuating subjective poverty across generations. Studies have indicated that individual social capital is significantly and negatively correlated with the intergenerational transmission of poverty. As a result, the stock of livelihood capital serves as an observable indication of residents' subjective poverty. Families with higher levels of livelihood capital tend to exhibit greater livelihood adaptability and are less prone to subjective poverty. Conversely, families with lower livelihood capital have reduced livelihood adaptability and are more susceptible to subjective poverty. Livelihood strategies play a crucial role in achieving the family's livelihood goals by effectively combining and utilizing various forms of capital. Scholars generally agree that livelihood strategies, especially non-agricultural ones, significantly contribute to breaking the cycle of poverty and fostering sustainable development.

**Physical capital and subjective poverty.** The interpretation of poverty from both physical and economic perspectives has been a topic of enduring interest among researchers. Physical capital, encompassing the infrastructure and production tools necessary for sustaining livelihoods, constitutes a vital aspect of multidimensional poverty. It is predominantly reflected in financial deficiencies concerning income, consumption, access to food, clothing, housing, and transportation [17, 18]. Poverty is a comprehensive term that encompasses economic, social, and cultural backwardness resulting from low income, a lack of essential physical goods and services required for life, and limited opportunities and resources for development. Economic factors are the fundamental drivers of poverty, making economic growth a powerful force in reducing both income and non-income poverty. From this viewpoint, physical poverty denotes a state of "survival poverty," where individuals lack the income and resources necessary to meet basic life needs.

Hypothesis 2: The more abundant physical capital is, the more subjective poverty can be reduced.

**Human capital and subjective poverty.** Human capital refers to the skills, knowledge, employability, and health that individuals possess when pursuing various livelihood strategies and striving to achieve their livelihood goals. From a macro perspective, labor mobility facilitates the efficient spatial and regional allocation of labor and its associated components, thereby reducing the relative likelihood of falling into poverty [19]. At the micro level, human capital significantly influences individual employment, income, and career advancement, directly impacting their overall welfare. The absence of human capital, which includes education level, labor skills, and health status, not only indicates relative poverty but also serves as a fundamental cause of such poverty. Enhancing the stock of human capital can play a pivotal role in helping the poor escape poverty [20, 21]. It can be asserted that human capital serves as an intrinsic strength for the sustainable development of impoverished families, fostering improvement in various aspects of livelihood capital. Amartya Sen has aptly pointed out that improved education and healthcare can directly enhance the quality of life and increase individuals' capacity to earn income, ultimately breaking the cycle of income poverty [22]. The provision of universal education and healthcare can significantly benefit the poor by offering them greater opportunities to overcome poverty.

Hypothesis 3: More sufficient human capital can reduce the occurrence of subjective poverty.

**Social capital and subjective poverty.** Social capital serves as an essential complement to traditional forms of capital, as it involves the characteristics of social organization that promote cooperative behavior, thereby enhancing societal efficiency [23]. At the individual level, social capital can occupy strategic network positions, allowing actors to acquire and utilize resources

embedded in their social relations, particularly those obtained from significant organizational positions within the social network. Social capital can be categorized into two types: structural social capital, which relates to social organizations, networks, and informal institutional arrangements, and cognitive social capital, encompassing various norms, trust, and cultural models prevalent in society. Numerous studies have underscored the significance of social capital, including social networks, trust, and norms, in poverty reduction. Social capital plays a crucial role in influencing family poverty and significantly impacts individuals' subjective perceptions of poverty [24, 25]. It can partially compensate for the limitations of human capital and financial capital. Moreover, there exists a mutually beneficial interactive relationship between different forms of capital. Physical and human capital aid individuals in expanding their social network, altering the nature of their relationships, and building a new type of social capital that facilitates the achievement of their goals. In turn, social capital acts as a lubricant, providing a conducive environment for optimal performance of physical and human capital.

Hypothesis 4: More sufficient social capital can reduce the occurrence of subjective poverty.

**Financial capital and subjective poverty.** Financial capital refers to the financial resources necessary for farmers to achieve their livelihood goals [26, 27]. It encompasses various wealth management products, including personal deposits in domestic and foreign currencies, treasury bonds, funds, securities, collective wealth, bank wealth management products, third-party depository deposits, insurance, gold and gold deposits, as well as collective fund trust plans and other wealth management products. Financial poverty, an essential aspect of multidimensional poverty, primarily manifests as a lack of sufficient funds. In response to the challenge of poverty, the Chinese government has introduced policies centered around the "trinity" big poverty-alleviation pattern, which emphasizes the integration of multiple strengths, measures, and support mechanisms. This approach involves special poverty alleviation initiatives, industrial poverty alleviation efforts, and social poverty alleviation programs. As such, the financial industry must play a pivotal role in fostering precise poverty alleviation, charting a hopeful path from poverty to prosperity, and contributing its financial strength to society's poverty alleviation endeavors. Poverty alleviation work extends beyond merely addressing basic needs like food and clothing; it also involves nurturing aspirations and wisdom. Financial poverty alleviation plays a significant role in comprehensive poverty alleviation efforts, focusing on cultivating the financial investment concepts of impoverished households to ensure multiple sources of economic income.

Hypothesis 5: The increase of sufficient financial capital can reduce the occurrence of subjective poverty.

## Livelihood strategies and subjective poverty

To achieve livelihood goals and attain positive livelihood outcomes, individuals employ livelihood strategies that involve combining and utilizing their owned livelihood assets. These strategies encompass a range of activities, such as production endeavors, investment plans, and fertility arrangements. The structure and quality of livelihood capital play a crucial role in determining the capacity of impoverished groups to adopt and implement livelihood strategies effectively. The tangible results of these strategies are evident in the benefits accrued from livelihood capital and the level of poverty risk faced by individuals. When individuals possess a higher level of diverse livelihood capital and employ various livelihood strategies, they can engage in both agricultural and non-agricultural activities, such as being chefs, welders,

cleaners, or pursuing other non-agricultural vocations. This increased diversity in livelihood activities strengthens their ability to withstand risks [28]. As a result, individuals with varying levels of livelihood capital and involved in diverse livelihood activities exhibit diverse outcomes, leading to the formation of differentiated livelihood strategies. The adoption of such diversified strategies is instrumental in reducing relative poverty. In essence, when an individual possesses a low total amount of livelihood capital and limited options for livelihood strategies, they may face a higher likelihood of subjective poverty. The ability to choose freely between multiple strategies becomes constrained in such circumstances, which may hinder their efforts to improve their living conditions and overall well-being.

Hypothesis 6: More diverse livelihood strategies can reduce the occurrence of subjective poverty.

## Data, variables and methods

### Data source

This paper employs data from the Chinese General Social Survey (CGSS), which the Renmin University of China implemented. The questionnaire covers basic personal information, family production and life, quality of life, social security, social participation, and other items. It adopts multistage stratified sampling, which is currently recognized as representative data with scientific research value in academia. To ensure the quality of the data and the accuracy of the statistical analysis, we eliminated samples with incomplete variables and some questions in the original answer, for example, "know," "not applicable," and "refuse to answer" were deemed invalid and then deleted. Finally, a valid sample size of 2,554 copies was obtained. The remaining sample size ensures that 30 provinces in China have a certain number of samples.

### Variables

**Dependent variable.** Various studies have explored different methods for measuring subjective poverty, aiming to bridge the gap between objective poverty measurements and the subjective experiences of residents [3] Since individuals have the most insightful information about their circumstances, they are best suited to judge whether they are in a state of poverty [1]. Researchers have identified two primary approaches to measuring subjective poverty based on its definition. The first approach involves collecting residents' subjective information and quantifying the data using specific questionnaires. Respondents are asked about their minimum living standards, which helps establish a subjective income poverty line for defining subjective poverty. However, this methodology often limits poverty measurement to income indicators, leading to a one-sided assessment [21]. The second approach is based on the "self-rating scale," inspired by the work of psychologist Cantril. This method gauges residents' satisfaction with their economic life using a graded measurement system [26]. Residents' self-determined evaluation results are then used to assess poverty levels among individuals. The results not only measure poverty and subjective well-being but also reflect residents' overall feelings about their living conditions and happiness. This second approach has become the mainstream method in empirical analysis. In this paper, we adopt the second approach for measuring subjective poverty. Specifically, we use the question "Which level do you think your socioeconomic status belongs to in the local area?" as the basis for measurement. The questionnaire offers a range of responses, including "up," "middle up," "middle," "middle down," and "down," which are graded with values of 1 to 5, respectively. Among these, "down" is identified as the indicator of subjective poverty and is assigned a value of 1, while the other responses are assigned a value of 2.

**Independent variables.** The core of public services supply lies in the authoritative and legal allocation of resources. The unequal distribution of resources and elements among various regions has exacerbated the occurrence of poverty among individuals. Evaluating the effectiveness of public services supply can be observed through individual behavioral choices under subjective consciousness, such as institutional trust and policy cognition. In this study, the public service satisfaction indicator is used to measure the status of the system's supply. It is important to note that the CGSS questionnaire measures individuals' subjective evaluation of the government's overall satisfaction with providing public services across 14 different aspects. These aspects include medical and health services, social security services, environmental protection, safeguarding political rights, maintaining social security, combating corruption, ensuring fairness in law enforcement, promoting economic development, expanding employment opportunities, providing open government information, improving service consciousness, safeguarding education equity, ensuring food and drug safety, and overall work and services. To ensure the reliability of the measurements, this paper conducts a reliability analysis. According to Table 1, the results show that the alpha value was 0.9325, and the alpha coefficient was greater than 0.6. It can be seen that this group of items has high reliability. We use the average satisfaction of the 14 aspects to measure public services supply status.

Physical capital plays a crucial role as a fundamental basis for poverty alleviation among the poor. It includes essential components such as infrastructure and production tools that fulfill their livelihood needs. However, due to the lack of viable investment channels and the limitations of their own capabilities, converting funds into physical capital poses significant challenges for the poor. In light of this, this article employs the question "Do you own real estate?" as a measure of the state of physical capital.

Financial capital represents the financial resources that individuals possess to pursue their life goals. As the economic foundation determines the overall superstructure, financial capital wields a decisive influence on residents' ability to thrive. Hence, it becomes imperative to proactively explore the involvement of financial funds in poverty alleviation and development, leveraging the potential impact of financial capital. In this study, "family income" is utilized as a measure of the endowment of financial capital.

Among various forms of livelihood capital, human capital holds a more central and crucial role. The human capital perspective primarily emphasizes the influence of economically valuable knowledge, skills, and physical abilities possessed by individuals in poverty. In this study, "education degree" is employed as the measure of human capital.

Social capital serves as a crucial complement to physical capital, encompassing the social resources and network of relationships that individuals can utilize in their livelihood activities. In this study, the question "What online social groups or circles have you joined in the past two years?" is employed as an indicator to measure social capital. It is important to note that the CGSS questionnaire includes the measurement of trust in various types of subjects. To ensure the reliability of the measurement, this article conducts a reliability analysis. The results from Table 2 reveal an alpha value of 0.6737, with the alpha coefficient exceeding 0.6, indicating high reliability for this group of items.

**Table 1. Alpha test.**

| Variable | Value |
|---|---|
| Average interitem covariance | 0.5767 |
| Number of items in the scale | 14 |
| Scale reliability coefficient | 0.9325 |

**Table 2. Alpha test.**

| Variable | Value |
|---|---|
| Average interitem covariance | 0.0188 |
| Number of items in the scale | 13 |
| Scale reliability coefficient | 0.6737 |

The livelihood strategy encompasses an individual's integrated utilization of livelihood capital, serving as a means of converting and leveraging the potential benefits of livelihood capital. The effectiveness of livelihood capital is closely tied to the adoption and implementation of livelihood strategies. When individuals have a wider range of diverse livelihood strategies at their disposal, it is more likely that their livelihood situation will improve. In this study, the "working way" is employed as an indicator to measure the choice of livelihood strategy and a detailed description of the variables can be found in Table 3.

## Model

Logistic regression analysis is mainly used to form a multiple linear regression relationship between a dependent variable and multiple independent variables and to predict the probability of an event. The advantage of logistic regression is that the independent variables can be continuous or discrete, and there is no need to satisfy a normal distribution, because in the general multivariate statistical analysis model, the variables must satisfy the normal distribution. In logistic regression analysis, the dependent variable $Y$ is a binary variable with values $Y = 1$ and $Y = 0$, representing subjective poverty and non-subjective poverty, respectively. The

**Table 3. Descriptive statistics of variables.**

| Variable name | Variable assignment | Mean | Std dev. | Min. | Max. |
|---|---|---|---|---|---|
| Subjective poverty | Poverty = 1<br>Non-poverty = 2 | 1.7941 | 0.4043 | 1 | 2 |
| Public service supply | Very bad = 1<br>Not very good = 2<br>Better = 3<br>Very good = 4 | 2.0125 | 0.5660 | 1 | 4 |
| Physical capital | Has not = 1<br>Has = 2 | 1.9415 | 0.2345 | 1 | 2 |
| Financial capital | Continuous variables, logarithmic processing | 10.7564 | 1.2315 | 0 | 15.8949 |
| Human capital | Not attending school = 1<br>Primary school = 2<br>Junior high school = 3<br>High school = 4<br>Technical secondary school = 5<br>Vocational and Technical School = 6<br>College = 7<br>Undergraduate college = 8<br>Graduate student = 9 | 3.9453 | 2.1577 | 1 | 9 |
| Social capital | No = 0<br>Yes = 1 | 0.3124 | 0.1512 | 0 | 0.8461 |
| Livelihood strategy | Engaged in nonagricultural work = 1<br>Mainly engaged in nonagricultural work while farming = 2<br>Mainly engaged in agriculture but also engaged in nonagricultural work = 3<br>Currently only farming = 4 | 2.1771 | 1.3582 | 1 | 4 |
| Region | Urban area = 1<br>Rural area = 2 | 1.4210 | 0.4937 | 1 | 2 |

$n$ independent variables that affect the value of $Y$ are $|X_1, X_2, \cdots$, and $X_n$, and the conditional probability of subjective poverty occurrence under the action of $n$ independent variables is $P = P(Y = 1|X_1, X_2, \cdots, X_n)$. The logistic regression model can be expressed as follows:

$$Z_i = a_o + a_1 X_{i1} \cdot + a_2 X_{i2} + \cdots + a_n X_{\text{in}} \tag{1}$$

$$P_i = \frac{1}{1 + exp(-z_i)} \tag{2}$$

$Z_i$ refers to the intermediate variable parameters, $a_0$ refers to the regression constants, $a_1$ refers the regression coefficient of the $j$ variable $(i, j, \cdots, n)$, $X_{ij}$ express the value of the i unit, and the value of the $j$ variable, if subjective poverty exists a value of 1, otherwise a value of 0, and $P_i$ refers the regression-prediction value of subjective poverty's probability of occurrence in the $i$ variable, $i = (1, 2, \cdots, n)$.

## Empirical results and analysis

### Analysis of spatial results

Before conducting the empirical analysis, we aim to visually present the provincial spatial characteristics of subjective poverty incidence, public service supply, and livelihood capital in China for the year 2019. To achieve this, we employ ArcMap to create a spatial distribution map that covers all 30 provinces in the country. To classify the variable values, we utilize the natural breakpoint hierarchy, which divides each variable into three categories: high-value area, medium-value area, and low-value area. For the incidence of subjective poverty, areas with a higher incidence represent a higher probability of subjective poverty occurrence. Regarding public service supply, higher values indicate that residents are less satisfied with the public services provided by the government. For physical capital, a higher value represents more house ownership. For financial capital, a higher value signifies higher household income. For human capital, a higher value denotes higher educational attainment. Finally, for social capital, a higher value indicates stronger social connections.

Based on the findings presented in Fig 1, it is evident that the high-value areas of subjective poverty incidence in China in 2019 were primarily concentrated in Heilongjiang, Jilin, and Shanxi. Meanwhile, Hebei, Chongqing, and Yunnan were categorized as medium-value areas in terms of subjective poverty incidence. On the other hand, the low-value areas, indicating a lower incidence of subjective poverty, were primarily observed in Fujian, Guangdong, Hubei, and Shanghai.

Based on the information presented in Fig 2, it is evident that the high-value areas of public service supply in China in 2019 were predominantly found in Liaoning, Jilin, Inner Mongolia, and Shanxi. Meanwhile, Hebei, Henan, Chongqing, and Yunnan were categorized as medium-value areas in terms of public service supply. On the other hand, the low-value areas, indicating lower levels of public service supply, were observed in Tibet and Qinghai.

Based on Fig 3, we discover that Jiangsu, Zhejiang, and Inner Mongolia were the high-value areas of physical capital in China in 2019. The medium-value areas mainly occurred in Henan, Hebei, Chongqing, and Yunnan. The low-value areas primarily occurred in Heilongjiang, Liaoning, and Ningxia.

From Fig 4, we know that the high-value areas of financial capital in China in 2019 mainly occurred in Jiangsu, Zhejiang, Anhui, and Shanghai. The medium-value areas occurred in Henan, Hebei, Chongqing, and Yunnan. Jilin, Gansu, and Tibet were low-value areas.

Based on the data presented in Fig 5, it is evident that Fujian, Zhejiang, and Shanghai were identified as the high-value areas of human capital in China in 2019. Meanwhile, the medium-

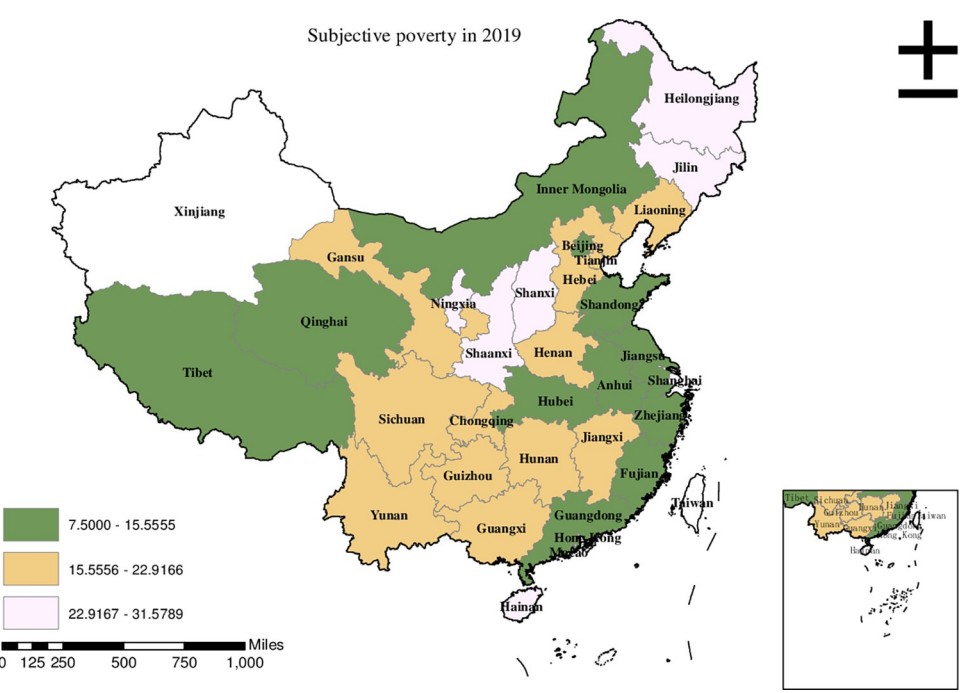

**Fig 1. Provincial spatial distribution of subjective poverty in 2019.**

value areas were primarily observed in Henan, Hebei, Chongqing, and Yunnan. On the other hand, the low-value areas, indicating lower levels of human capital, were predominantly found in Liaoning, Qinghai, and Tibet.

Based on the data presented in Fig 6, it is evident that the high-value areas of social capital in China in 2019 were primarily concentrated in Jiangsu, Zhejiang, Fujian, and Shanghai. Meanwhile, the medium-value areas were observed in Henan, Hebei, Heilongjiang, and Shandong. On the other hand, the low-value areas, indicating lower levels of social capital, were predominantly found in regions such as Tibet.

Upon comparing Figs 1–6, we have observed a significant overlap between the high, medium, and low-value areas of subjective poverty incidence and the high, medium, and low-value areas of public service supply and the four types of capital. Provinces and cities with higher subjective poverty incidence also tend to exhibit higher levels of public service dissatisfaction among residents and lower ownership of all four types of capital. These findings suggest a provincial spatial connection between subjective poverty incidence and public service supply, as well as physical capital, financial capital, human capital, and social capital.

## Correlation analysis

Table 4 shows the correlation analysis of the Pearson correlation analysis results between the variables.

The variable of subjective poverty is found to be significantly correlated with the variable of public services supply, as well as with physical capital, financial capital, human capital, social capital, and livelihood strategies, all at the 1% significance level. This preliminary analysis suggests that higher levels of public services supply and more diverse livelihood strategies are associated with a lower likelihood of subjective poverty. However, it should be noted that the variables of physical capital, financial capital, and social capital did not show significant correlations with the other variables. Apart from physical capital, financial capital, and social capital,

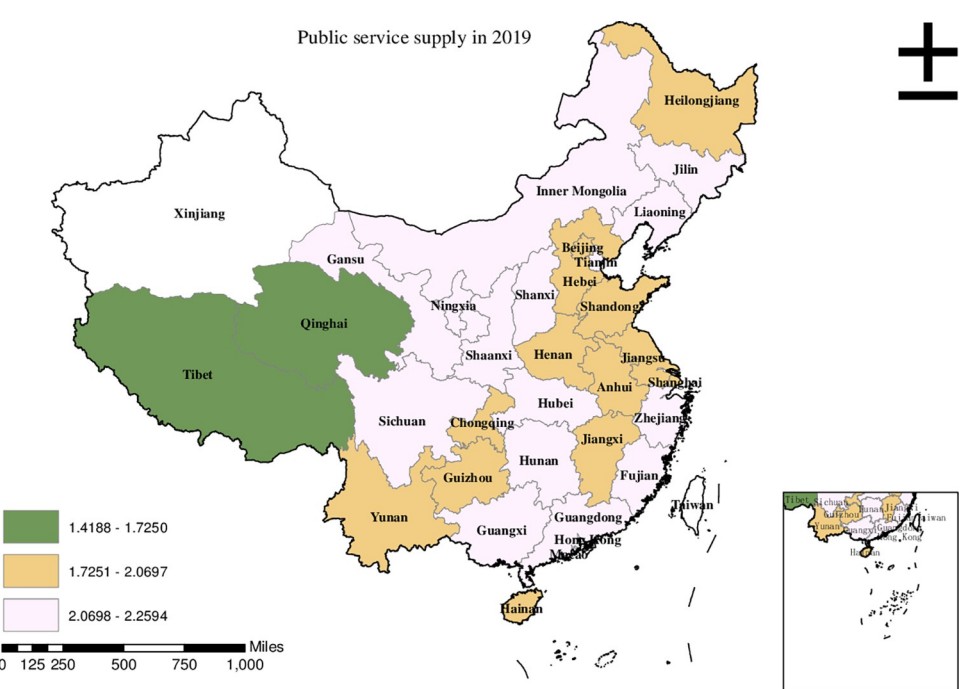

**Fig 2. Provincial spatial distribution of public service supply in 2019.**

all other variables demonstrated either positive or negative correlations at the 1% or 5% significance level. This indicates that they may have a strong influence on the occurrence of subjective poverty. To gain a deeper understanding of these relationships, regression analysis will be

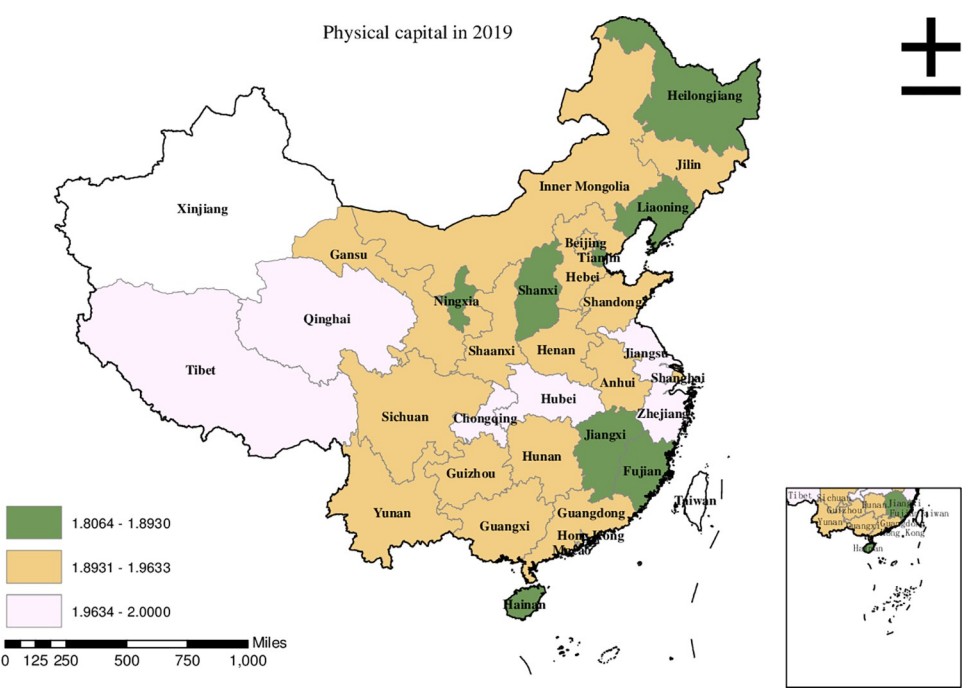

**Fig 3. Provincial spatial distribution of physical capital in 2019.**

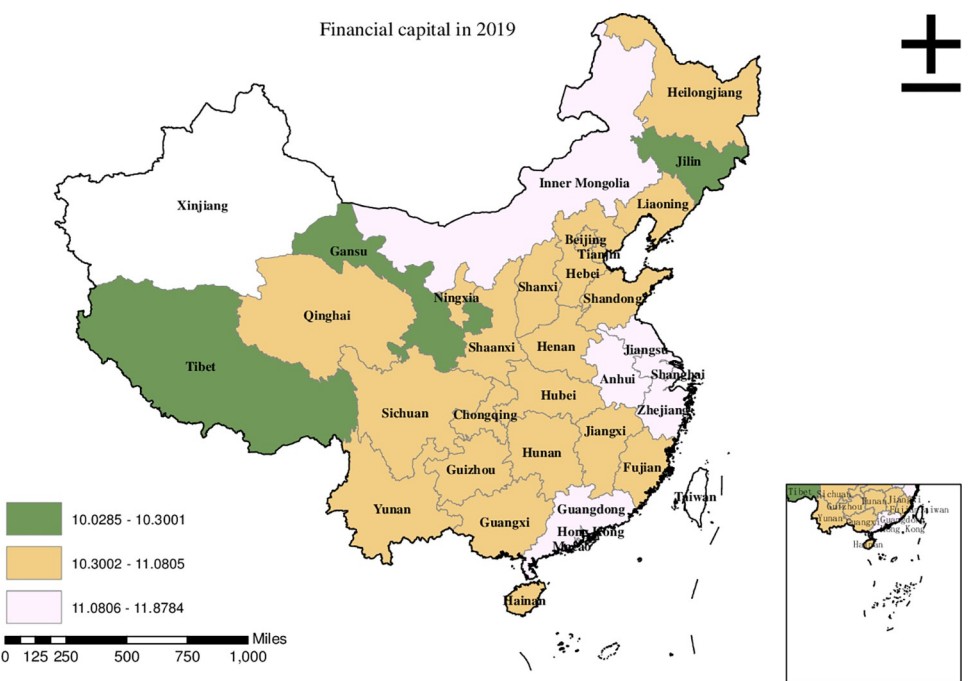

**Fig 4. Provincial spatial distribution of financial capital in 2019.**

conducted on each variable. This analysis aims to explore the specific impacts of each factor on subjective poverty and identify potential areas for policy interventions and poverty alleviation efforts. By investigating these correlations, we can better comprehend the complex dynamics

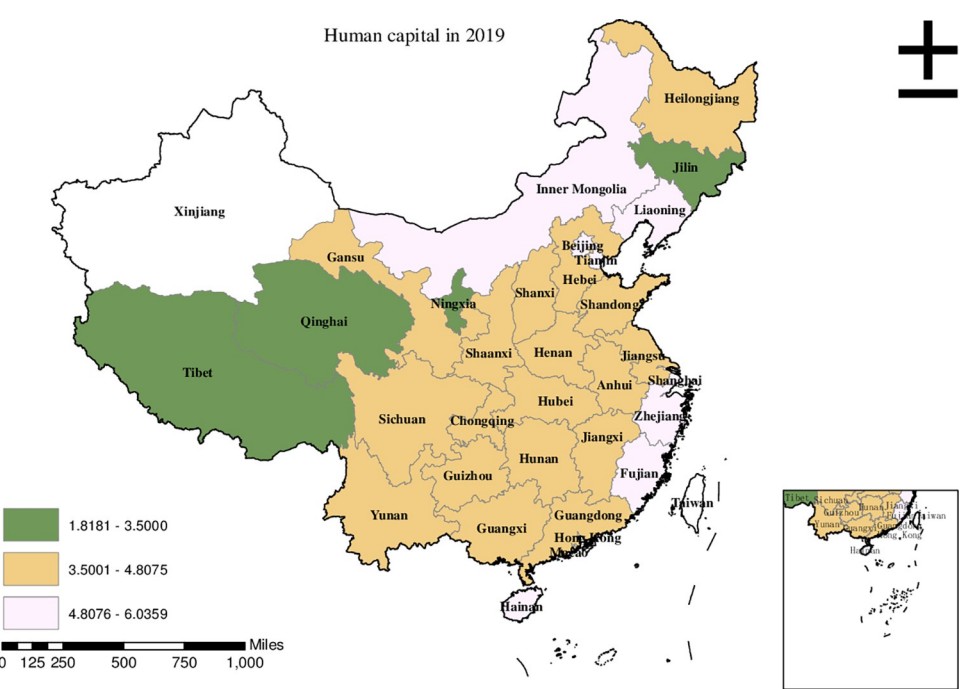

**Fig 5. Provincial spatial distribution of human capital in 2019.**

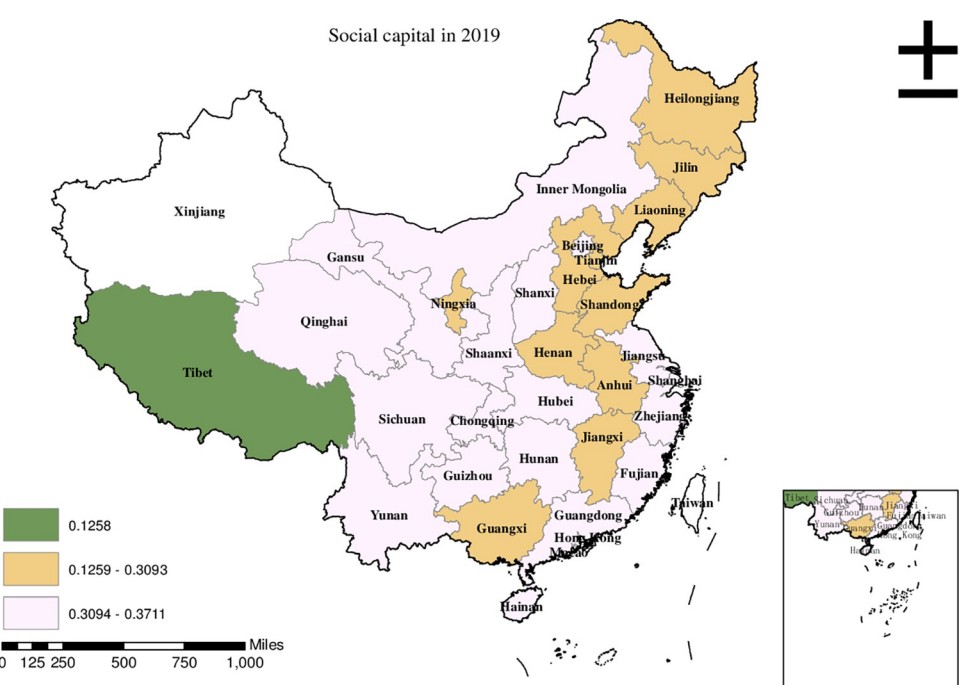

**Fig 6. Provincial spatial distribution of social capital in 2019.**

of subjective poverty and contribute to the development of effective strategies to combat poverty and improve the well-being of residents in different regions of China.

## Benchmark regression

To examine the presence of collinearity in the multiple linear regression model variables, we conducted a Variance Inflation Factor test. The test results revealed that the average VIF value was 1.33, which is below the critical value of 10. This indicates that there is no collinearity issue among the independent variables.

Column (1) of Table 5 examines the relationship between subjective poverty and public services supply. The results indicate a significant negative effect of public services supply on the

**Table 4. Pearson correlation analysis of variables.**

| VARIABLES | Subjective poverty | Public service supply | Physical capital | Financial capital | Human capital | Social capital | Livelihood strategy |
|---|---|---|---|---|---|---|---|
| Subjective poverty | 1.0000*** | | | | | | |
| Public service supply | -0.0812*** | 1.0000*** | | | | | |
| Physical capital | 0.1012*** | -0.0227 | 1.0000*** | | | | |
| Financial capital | 0.1491*** | 0.0198 | 0.0680** | 1.0000*** | | | |
| Human capital | 0.1654*** | -0.0347** | 0.0027 | 0.3703*** | 1.0000*** | | |
| Social capital | 0.0985*** | -0.0104*** | 0.0335** | 0.2604 | 0.2964*** | 1.0000*** | |
| Livelihood strategy | -0.0522*** | -0.0018 | 0.00759*** | -0.3663*** | -0.4524*** | -0.2834*** | 1.0000*** |

Standard errors in parentheses

*** p<0.01

** p<0.05

* p<0.1.

**Table 5. Estimated results of the logit model.**

| VARIABLES | (1) | (2) | (3) | (4) | (5) | (6) |
|---|---|---|---|---|---|---|
| | | | | | urban | rural |
| Public service supply | -0.0569*** | | | -0.0398*** | -0.0397*** | -0.0374*** |
| | (0.0052) | | | (0.0085) | (0.0110) | (0.0137) |
| Physical capital | | | 0.1740*** | 0.1490*** | 0.1540*** | 0.1160* |
| | | | (0.0206) | (0.0268) | (0.0294) | (0.0648) |
| Financial capital | | | 0.0343*** | 0.0325*** | 0.0240*** | 0.0514*** |
| | | | (0.0045) | (0.00574) | (0.00691) | (0.0103) |
| Human capital | | | 0.0169*** | 0.0244*** | 0.0274*** | 0.0218*** |
| | | | (0.0025) | (0.0032) | (0.0037) | (0.0070) |
| Social capital | | | 0.1030*** | 0.1070*** | 0.1150** | 0.0898 |
| | | | (0.0349) | (0.0411) | (0.0508) | (0.0696) |
| Livelihood strategy | | -0.0273*** | | 0.0160*** | 0.0128 | 0.0086 |
| | | (0.0036) | | (0.0056) | (0.0098) | (0.0088) |
| Constant | 1.901*** | 1.843*** | 0.9900*** | 1.089*** | 1.153*** | 0.998*** |
| | (0.0127) | (0.0098) | (0.0599) | (0.0817) | (0.0940) | (0.1750) |
| Observations | 7,390 | 6,567 | 5,963 | 3,952 | 2,461 | 1,491 |
| R-squared | 0.0114 | 0.0083 | 0.0436 | 0.0535 | 0.0646 | 0.0427 |

Standard errors in parentheses

*** $p < 0.01$

** $p < 0.05$

* $p < 0.1$.

occurrence of subjective poverty at the 1% significance level, confirming the validity of Hypothesis 1. This suggests that a well-functioning and comprehensive government public services supply can effectively reduce the occurrence of subjective poverty. In Column (2), there is a negative correlation between livelihood strategies and the occurrence of subjective poverty, supporting the validity of Hypothesis 6. The findings demonstrate that diverse livelihood strategies, encompassing both agricultural and non-agricultural activities, can lead to increased income and a reduction in subjective poverty. Column (3) investigates the relationship between capital conversion and subjective poverty. It reveals that physical capital, financial capital, human capital, and social capital all have significant positive effects on subjective poverty at the 1% significance level, thus confirming the establishment of Hypotheses 2–5. The greater the abundance of physical capital, financial capital, human capital, and social capital, the more subjective poverty occurrence can be reduced. Furthermore, Column (4) presents the regression results after accounting for the control variables, and it verifies the robustness of the research findings. The conclusions drawn in the previous columns remain valid, indicating the consistency and reliability of the results.

As presented in Table 5, Columns (5) and (6) present the regression results after controlling for urban-rural attributes. The findings reveal that public services supply, physical capital, financial capital, and human capital all have varying degrees of impact on the occurrence of subjective poverty among urban and rural residents at different levels of significance. These results are consistent with previous research findings by other scholars [20]. Specifically, public services supply, human capital, and physical capital have a more pronounced effect on the subjective poverty of urban residents compared to rural residents. On the other hand, financial capital has a greater impact on the subjective poverty of rural residents than urban residents. Additionally, social capital solely affects the occurrence of subjective poverty among urban

residents. These observations suggest that the factors influencing subjective poverty among urban residents are more diverse and multifaceted, whereas rural communities are still heavily influenced by economic factors. It indicates that rural groups possess significantly lower economic capacity compared to their urban counterparts and face challenges related to weak social capital. This discrepancy may be attributed to the urban-rural divide in terms of economic development, access to resources, and opportunities for social connectivity.

## Robustness test

**Replaced the dependent variable.** To ensure the robustness of our empirical findings, we employed a different variable as the explained variable in our analysis. We used the question "How to evaluate your family's economic situation?" from the questionnaire, with respondents providing a value on a scale from 1 to 10 to grade their family's economic situation. We then processed the data on subjective poverty accordingly and reassigned values to create a new variable. We categorized respondents who rated their family's economic situation as less than 5 points as experiencing subjective poverty, assigning them a value of 1 while assigning a value of 2 to the rest. The results obtained from this new analysis, as shown in Table 6, were found to be consistent with those in Table 5. Public services supply, physical capital, financial capital, human capital, social capital, and livelihood strategies still demonstrated significant effects on the occurrence of subjective poverty at varying levels of significance. This provides further evidence to support the validity and reliability of our model.

**Change model.** To further validate the robustness of our empirical findings, we employed the Generalized Linear Model (GLM) as an alternative statistical approach, and the results are presented in Table 7. We are pleased to report that the outcomes obtained through the GLM model are consistent with those derived from the logistic model. Once again, we observe that

**Table 6. Robustness test—change variable.**

| VARIABLES | (1) | (2) | (3) | (4) |
|---|---|---|---|---|
| *Public service supply* | -0.0573*** | | | -0.0379*** |
| | (0.00484) | | | (0.00769) |
| *Physical capital* | | 0.0988*** | | 0.0920*** |
| | | (0.0183) | | (0.0241) |
| *Financial capital* | | 0.0463*** | | 0.0369*** |
| | | (0.00407) | | (0.00515) |
| *Human capital* | | 0.0242*** | | 0.0208*** |
| | | (0.00226) | | (0.00292) |
| *Social capital* | | 0.0407 | | 0.0130 |
| | | (0.0310) | | (0.0369) |
| *Livelihood strategy* | | | -0.0522*** | -0.0248*** |
| | | | (0.00337) | (0.00510) |
| Constant | 1.950*** | 1.037*** | 1.939*** | 1.307*** |
| | (0.0116) | (0.0532) | (0.00901) | (0.0733) |
| Observations | 7,390 | 5,963 | 6,567 | 3,952 |
| R-squared | 0.0137 | 0.0673 | 0.0352 | 0.0802 |

Standard errors in parentheses

*** $p<0.01$

** $p<0.05$

* $p<0.1$.

**Table 7. Robustness test—GLM model.**

| VARIABLES | (1) | (2) | (3) | (4) |
|---|---|---|---|---|
| *Public service supply* | -0.0569*** | | | -0.0397*** |
| | (0.0052) | | | (0.0085) |
| *Physical capital* | | 0.1741*** | | 0.1494*** |
| | | (0.0206) | | (0.0268) |
| *Financial capital* | | 0.0343*** | | 0.0325*** |
| | | (0.0045) | | (0.0057) |
| *Human capital* | | 0.0168*** | | 0.0244*** |
| | | (0.0025) | | (0.0032) |
| *Social capital* | | 0.1032 | | 0.1066** |
| | | (0.0348) | | (0.0410) |
| *Livelihood strategy* | | | -0.0272*** | -0.0159*** |
| | | | (0.0036) | (0.0816) |
| Constant | 1.9013*** | 0.9901*** | 1.8428*** | 1.0890*** |
| | (0.0127) | (0.05989) | (0.0098) | (0.0816) |
| Observations | 7,390 | 5,963 | 6,567 | 3,952 |
| AIC | 1.0902 | 0.9309 | 1.0661 | 0.8803 |

Standard errors in parentheses

*** $p < 0.01$

** $p < 0.05$

* $p < 0.1$.

public services supply, physical capital, financial capital, human capital, social capital, and livelihood strategy continue to exert significant effects on the occurrence of subjective poverty. The consistency between the results obtained from both the logistic and GLM models reinforces the credibility and reliability of our findings. It indicates that the impact of public services supply and various livelihood capitals on subjective poverty is robust and reliable across different statistical techniques. This strengthens the validity of our research and bolsters confidence in the relationships identified.

## Research conclusions and policy implications

### Research conclusions

This study highlights the multidimensional nature of poverty, where economic poverty is just one aspect. Addressing poverty requires not only tackling economic challenges but also addressing the psychological and subjective aspects of poverty. Understanding the influencing factors of subjective poverty is crucial for the government to adjust policies effectively and promote social and economic reforms. The results demonstrate that public services supply has a significant negative effect on the occurrence of subjective poverty at the 1% significance level. Additionally, physical capital, financial capital, human capital, social capital, and livelihood strategy all have significant positive effects on the occurrence of subjective poverty at the 1% significance level. Specifically, for each one-unit change in public services supply, the probability of subjective poverty occurrence decreases by 0.0398. For each one-unit change in physical capital and financial capital, the probability of subjective poverty occurrence decreases by 0.149 and 0.0325, respectively. Similarly, for each one-unit change in human capital and social capital, the probability of subjective poverty occurrence decreases by 0.0244 and 0.107, respectively. The formation of subjective poverty is influenced by a combination of multiple factors,

and these factors have different effects on urban and rural areas. The results indicate that public services supply, physical capital, financial capital, and human capital affect the occurrence of subjective poverty among both urban and rural residents at varying significance levels. However, social capital only affects the occurrence of subjective poverty among urban residents.

## Policy implications

Governments should prioritize supply-side reforms and enhance the provision of public services to tackle poverty effectively. A key aspect is to reform the income distribution system and the fiscal and taxation system, ensuring a fair distribution of income. This entails strengthening income distribution controls and implementing policies to create a redistributive adjustment mechanism that utilizes taxation and social security measures, with a greater focus on directing benefits toward the poor. Fiscal policy should play a crucial role in supporting impoverished regions by increasing financial assistance and encouraging financial institutions to provide more credit to these areas, addressing the issue of limited capital accumulation. Coordination between social security and poverty reduction policies is paramount. Social security measures should be leveraged to alleviate subjective poverty, and a sustainable mechanism should be established to consolidate the achievements of poverty alleviation efforts. Additionally, policymakers must prioritize the elimination of subjective poverty and enhancing residents' well-being as key objectives of social security policies. To achieve these objectives, the government should adopt a rational time frame and spatial planning to effectively address subjective poverty in different regions and stages. Implementing pilot programs can help identify successful models, which can then be expanded gradually for broader impact.

Governments must prioritize capital conversion as a key strategy in their poverty reduction efforts. Ensuring equal access to education should be a top priority, and this can be achieved by strengthening educational opportunities in impoverished regions and low-income families. To do so, the government can implement financial discounts and offer guarantees to encourage financial institutions to support education and increase targeted enrollment in underprivileged areas. Additionally, improving educational infrastructure and providing high-quality resources in areas with educational deficits will be crucial to leveling the playing field. Furthermore, vocational skills training should be normalized in impoverished areas, and the government should offer free vocational skills education to the poor. By equipping individuals with the necessary skills and knowledge, the government can empower them to become self-sufficient and break free from the cycle of poverty. Regulating the real estate market to boost investment in affordable housing is also essential. This will allow citizens to access education, employment, medical care, and housing, ultimately improving their overall quality of life. By enhancing the living conditions of the poor and providing them with access to essential services, governments can effectively reduce poverty and promote the well-being of all their citizens. These efforts will create a more inclusive and equitable society where everyone has the opportunity to thrive.

Based on our findings, it is evident that the nature of work may not directly impact subjective poverty. However, employment remains a critical factor affecting income, which, in turn, influences subjective poverty. To address this issue, governments should prioritize the improvement of labor market institutions to enhance employment opportunities and participation rates. Moreover, governments should explore the potential of "Internet + livelihood strategies" to leverage modern information technology and the development of the tertiary industry. By doing so, they can expand livelihood strategies and options, stimulating economic

opportunities for groups vulnerable to subjective poverty. Adopting a localized approach is also crucial for governments. By capitalizing on the unique strengths and resources of each area, they can develop characteristic industries and promote employment in those regions.

## Supporting information

**S1 Data.**
(ZIP)

## Acknowledgments

Special thanks for the advice and assistance from anonymous experts.

## Limitation of the study

The data is missing on the natural capital variable in this paper, and we could not discuss the effect of natural capital on subjective poverty. Therefore, missingness in the data is the main limitation of the study.

## Author Contributions

**Data curation:** Li Chen, Yuan Meng.

**Formal analysis:** Yuanquan Lu, Li Chen, Yuan Meng.

**Methodology:** Li Chen, Yuan Meng.

**Software:** Yuan Meng.

**Supervision:** Yuanquan Lu.

**Writing – original draft:** Yuanquan Lu, Li Chen.

**Writing – review & editing:** Yuanquan Lu, Li Chen.

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
