## [Decision Letter · Decision Letter 0]

27 Jun 2023

PONE-D-23-13234How do public services supply, livelihood capital, and livelihood strategies affect subjective poverty?PLOS ONE

Dear Dr. Lu,

Thank you for submitting your manuscript to PLOS ONE. After careful consideration, we feel that it has merit but does not fully meet PLOS ONE’s publication criteria as it currently stands. Therefore, we invite you to submit a revised version of the manuscript that addresses the points raised during the review process. Please submit your revised manuscript by Aug 11 2023 11:59PM. If you will need more time than this to complete your revisions, please reply to this message or contact the journal office at plosone@plos.org. Please include the following items when submitting your revised manuscript:A rebuttal letter that responds to each point raised by the academic editor and reviewer(s). You should upload this letter as a separate file labeled 'Response to Reviewers'.A marked-up copy of your manuscript that highlights changes made to the original version. You should upload this as a separate file labeled 'Revised Manuscript with Track Changes'.An unmarked version of your revised paper without tracked changes. You should upload this as a separate file labeled 'Manuscript'.

We look forward to receiving your revised manuscript.

Kind regards,

C. A. Zúniga-González, Ph.D

Academic Editor

PLOS ONE

Journal Requirements:

3. We note that Figures 1 to 6 in your submission contain map images which may be copyrighted. All PLOS content is published under the Creative Commons Attribution License (CC BY 4.0), which means that the manuscript, images, and Supporting Information files will be freely available online, and any third party is permitted to access, download, copy, distribute, and use these materials in any way, even commercially, with proper attribution. For these reasons, we cannot publish previously copyrighted maps or satellite images created using proprietary data, such as Google software (Google Maps, Street View, and Earth). For more information, see our copyright guidelines: http://journals.plos.org/plosone/s/licenses-and-copyright.

(1) You may seek permission from the original copyright holder of Figures 1 to 6 to publish the content specifically under the CC BY 4.0 license.  

**Additional Editor Comments:**

Dear authors, I am checked the observations of the reviewer. My decision is to make minor changes. I suggest empathizing with the significant of this study, improve the methodology section addressing the statistical processing data, it is very important added recently references comparting with your results.

Reviewers' comments:

Reviewer's Responses to Questions

**Comments to the Author**

1. Is the manuscript technically sound, and do the data support the conclusions?

Reviewer #1: Partly

Reviewer #2: Yes

2. Has the statistical analysis been performed appropriately and rigorously? 

Reviewer #1: Yes

Reviewer #2: Yes

3. Have the authors made all data underlying the findings in their manuscript fully available?

Reviewer #1: Yes

Reviewer #2: Yes

4. Is the manuscript presented in an intelligible fashion and written in standard English?

Reviewer #1: No

Reviewer #2: Yes

5. Review Comments to the Author

Reviewer #1: Follow the review report strictly, the manuscript could benefit from a final proofreading to correct minor typographical errors and ensure consistency in formatting and citation style. I suggest to get help from a professionals.

Reviewer #2: The paper has a high degree of actuality and relevance as it deals with the topic of poverty, not only from an economic perspective but also as a social problem and poses the question about certain limitations as objective poverty based on the population’s income does not reflect also important subjective aspects, with regard to feelings, education opportunities, pension, medical security and participation in decision-making of the affected individuals.

The introduction substantiates the approach of the paper and shows how poverty is a shared issue for the field of economic and social studies, as objective measurements have limitations and fail to capture some important subjective aspects related to perceptions and state-of-affairs generally related to the well-being of individuals. Hence, the authors of the paper attempt to shift the focus on this study dealing with Chinese poverty also to the subjective dimension next to the objective one, by analyzing how public services supply, livelihood capital, and livelihood strategies correlate significantly with subjective poverty. Moreover, in approaching the topic, the authors make interesting observations about studies concerned with studying the objective poverty line, referring to the Dutch scholars who firs pioneered the concept of subjective poverty line measurement.

The literature review is comprehensive consisting of a sound enumeration of the main results obtained by different scholars from various countries regarding the study of the objective and subjective poverty line, and the developments underwent by these concepts in time, along with some pertinent results obtained by the other studies.

The theoretical basis and the research hypothesis are well substantiated, and comprehensively detailed, while the formulated hypotheses are sound and well formulated in explaining the possible relationships and foreseen outcomes of the respective correlations between public services supply and subjective poverty, livelihood capital and subjective poverty, physical capital and subjective poverty, human capital and subjective poverty, and social capital and subjective poverty, etc.

The methodological approach is built on sound data source collection and processing, and the selected variables are fit for the purposes of the paper, while the methods employed contribute to demonstrating the pursued goals according to the hypotheses.

The empirical results and their analysis, of both spatial results and of the considered correlations and regressions are convincing and provide for obtaining a clear image in relation with the topic dealt with in the paper.

Research conclusions and policy implications underpin, based on the formulated hypotheses, and the employed mathematical-statistical analyses, the fact that poverty is a multidimensional problem, and that it should be dealt with not only from economic, but also from the social perspective which allows for considerations of psychological, implicitly subjective factors to be introduced in attempting to identify ways of dealing also with the subjective side of poverty. Hence, the authors provide an interesting solution that may be taken into account by decision factors, and government in adjusting policies timely and in promoting the necessary social and economic reforms.

Even though the study is performed for the Chinese context, it might serve as reference for considering policy implications not only for the Chinese context, but also for other countries of the world, that might contribute to improving their own results regarding income distribution systems, fiscal and taxation systems for promoting a fairer distribution of incomes.

The paper is soundly built, and the approach has originality, providing for an interesting perspective which is necessary in the current uncertain economic and social environment. We recommend only a rereading for improving minor English language errors. The paper is recommended for publication as it is a valuable contribution, and relevant with respect to the stated goal which is fully achieved.

6. PLOS authors have the option to publish the peer review history of their article (what does this mean?). If published, this will include your full peer review and any attached files.

Reviewer #1: No

Reviewer #2: **Yes: **Laura-Mariana Cismaș

---

## [Author Response · Author response to Decision Letter 0]

3 Aug 2023

Detailed response to reviewers

Reference: PONE-D-23-13234

Title: How do public services supply, livelihood capital, and livelihood strategies affect subjective poverty?

Journal title: PLOS ONE

Authors: Li Chen, Yuanquan Lu, Yuan Meng

Dear Editors and Reviewers,

Thanks for your letter and comments on our manuscript titled "How do public services supply, livelihood capital, and livelihood strategies affect subjective poverty?" (PONE-D-23-13234). These comments helped us improve our manuscript and provided important guidance for future research.

We have addressed the editor's and the reviewers' comments to the best of our abilities and revised the text to meet the PLOS ONE style requirements. We hope this meets your requirements for a publication.

We marked the revised portions in red in the revised manuscript with track changes. The main comments and our specific responses are detailed below:

Editors:

Dear authors, I am checked the observations of the reviewer. My decision is to make minor changes. I suggest empathizing with the significant of this study, improve the methodology section addressing the statistical processing data, it is very important added recently references comparting with your results.

Response:

Thank you for the editor's suggestions. In the revised manuscript, we have added some latest research and removed some older studies. The specific modifications are as follows.

The economic aspect mainly involves the relationship between subjective poverty and factors such as socio-economic background, income level, household disposable income, and employment status[9]. It has been found that all of these factors significantly affect the occurrence of subjective poverty directly or indirectly[1,3]. Joyce and Ziliak[10] compare the causes of relative poverty in Britain and the United States and find that relative poverty in Britain is related to working households, while low-skilled is the main cause of relative poverty in the United States. Some scholars have also studied the relationship between policy and poverty. Li and Li[11] find that the anti-poverty relocation and settlement program affects subjective well-being by affecting the probability that people will be exposed to risks due to policy. Agasisti et al.[12] collected relevant data from European countries between 2006 and 2015 and found a significant correlation between education and poverty. Zheng and Wang[13] discovered that social pension insurance can effectively reduce poverty among 95% of rural elderly individuals.

Reviewer #1:

Follow the review report strictly, the manuscript could benefit from a final proofreading to correct minor typographical errors and ensure consistency in formatting and citation style. I suggest to get help from a professionals.

Response:

Thank you for the expert's thorough review, and we greatly appreciate the expert's recognition of our research. In the revised manuscript, we carefully checked all the content and corrected any errors. Once again, thank you for the expert's feedback.

Reviewer #2: 

The paper has a high degree of actuality and relevance as it deals with the topic of poverty, not only from an economic perspective but also as a social problem and poses the question about certain limitations as objective poverty based on the population’s income does not reflect also important subjective aspects, with regard to feelings, education opportunities, pension, medical security and participation in decision-making of the affected individuals.

The introduction substantiates the approach of the paper and shows how poverty is a shared issue for the field of economic and social studies, as objective measurements have limitations and fail to capture some important subjective aspects related to perceptions and state-of-affairs generally related to the well-being of individuals. Hence, the authors of the paper attempt to shift the focus on this study dealing with Chinese poverty also to the subjective dimension next to the objective one, by analyzing how public services supply, livelihood capital, and livelihood strategies correlate significantly with subjective poverty. Moreover, in approaching the topic, the authors make interesting observations about studies concerned with studying the objective poverty line, referring to the Dutch scholars who firs pioneered the concept of subjective poverty line measurement.

The literature review is comprehensive consisting of a sound enumeration of the main results obtained by different scholars from various countries regarding the study of the objective and subjective poverty line, and the developments underwent by these concepts in time, along with some pertinent results obtained by the other studies.

The theoretical basis and the research hypothesis are well substantiated, and comprehensively detailed, while the formulated hypotheses are sound and well formulated in explaining the possible relationships and foreseen outcomes of the respective correlations between public services supply and subjective poverty, livelihood capital and subjective poverty, physical capital and subjective poverty, human capital and subjective poverty, and social capital and subjective poverty, etc.

The methodological approach is built on sound data source collection and processing, and the selected variables are fit for the purposes of the paper, while the methods employed contribute to demonstrating the pursued goals according to the hypotheses.

The empirical results and their analysis, of both spatial results and of the considered correlations and regressions are convincing and provide for obtaining a clear image in relation with the topic dealt with in the paper.

Research conclusions and policy implications underpin, based on the formulated hypotheses, and the employed mathematical-statistical analyses, the fact that poverty is a multidimensional problem, and that it should be dealt with not only from economic, but also from the social perspective which allows for considerations of psychological, implicitly subjective factors to be introduced in attempting to identify ways of dealing also with the subjective side of poverty. Hence, the authors provide an interesting solution that may be taken into account by decision factors, and government in adjusting policies timely and in promoting the necessary social and economic reforms.

Even though the study is performed for the Chinese context, it might serve as reference for considering policy implications not only for the Chinese context, but also for other countries of the world, that might contribute to improving their own results regarding income distribution systems, fiscal and taxation systems for promoting a fairer distribution of incomes.

The paper is soundly built, and the approach has originality, providing for an interesting perspective which is necessary in the current uncertain economic and social environment. We recommend only a rereading for improving minor English language errors. The paper is recommended for publication as it is a valuable contribution, and relevant with respect to the stated goal which is fully achieved.

Response:

Thank you very much for the expert's careful review, and we greatly appreciate the expert's recognition of our manuscript. In the original manuscript, there were indeed some language errors, which were our oversight, and we sincerely apologize for that. In the revised manuscript, we carefully examined the language throughout the entire article and corrected the language errors. Once again, thank you for the expert's feedback.

Thank you again for taking the time to review our manuscript and provide valuable feedback. We greatly appreciate your expertise and insightful comments, which have undoubtedly improved the quality of our research. We also extend our gratitude to the editors for their assistance throughout the review process. Thank you for your support and guidance. We look forward to hearing from you regarding our submission. We would be glad to respond to any further questions and comments that you may have.

Best wishes to you.

Kind regards,

Li Chen, Yuanquan Lu, Yuan Meng

---

## [Decision Letter · Decision Letter 1]

29 Aug 2023

PONE-D-23-13234R1How do public services supply, livelihood capital, and livelihood strategies affect subjective poverty?PLOS ONE

Dear Dr. Lu,

Thank you for submitting your manuscript to PLOS ONE. After careful consideration, we feel that it has merit but does not fully meet PLOS ONE’s publication criteria as it currently stands. Therefore, we invite you to submit a revised version of the manuscript that addresses the points raised during the review process.

We look forward to receiving your revised manuscript.

Kind regards,

Chaohai Shen

Academic Editor

PLOS ONE

Journal Requirements:

Reviewers' comments:

Reviewer's Responses to Questions

**Comments to the Author**

1. If the authors have adequately addressed your comments raised in a previous round of review and you feel that this manuscript is now acceptable for publication, you may indicate that here to bypass the “Comments to the Author” section, enter your conflict of interest statement in the “Confidential to Editor” section, and submit your "Accept" recommendation.

Reviewer #1: (No Response)

Reviewer #2: All comments have been addressed

2. Is the manuscript technically sound, and do the data support the conclusions?

Reviewer #1: Partly

Reviewer #2: Yes

3. Has the statistical analysis been performed appropriately and rigorously? 

Reviewer #1: Yes

Reviewer #2: Yes

4. Have the authors made all data underlying the findings in their manuscript fully available?

Reviewer #1: Yes

Reviewer #2: Yes

5. Is the manuscript presented in an intelligible fashion and written in standard English?

Reviewer #1: Yes

Reviewer #2: Yes

6. Review Comments to the Author

Reviewer #1: The paper is interesting and well-written overall. However, it may be enhanced by including more precise illustrations, critical evaluations, and discussions of the results.

Reviewer #2: The paper is recommended for publication as it is a valuable contribution, and relevant with respect to the stated goal which is fully achieved. I recommend the publication of this paper in its current form.

7. PLOS authors have the option to publish the peer review history of their article (what does this mean?). If published, this will include your full peer review and any attached files.

Reviewer #1: No

Reviewer #2: **Yes: **Laura-Mariana Cismaș

---

## [Author Response · Author response to Decision Letter 1]

11 Sep 2023

Detailed response to reviewers

Reference: PONE-D-23-13234R1

Title: How do public services supply, livelihood capital, and livelihood strategies affect subjective poverty?

Journal title: PLOS ONE

Authors:Yuanquan Lu, Li Chen, Yuan Meng

Dear Editors and Reviewers,

Thanks for your letter and comments on our manuscript titled "How do public services supply, livelihood capital, and livelihood strategies affect subjective poverty?" (PONE-D-23-13234R1). These comments helped us improve our manuscript and provided important guidance for future research.

We have addressed the editor's and the reviewers' comments to the best of our abilities and revised the text to meet the PLOS ONE style requirements. We hope this meets your requirements for a publication.

We marked the revised portions in red in the revised manuscript with track changes. The main comments and our specific responses are detailed below:

Editors:

Response:

Thank you for the editor's suggestions. In the revised manuscript. We have removed the citation of the withdrawn paper, which we consider to be of little significance. We have also revised the content of the body of the manuscript in accordance with the cited papers.

Reviewer #1:

The paper is interesting and well-written overall. However, it may be enhanced by including more precise illustrations, critical evaluations, and discussions of the results.

Response:

Thank you very much for your comments. In the original manuscript, we did have problems with unclear and incorrect corresponding images, so we are very sorry and thank the reviewers for their careful review. We have revised this part in the revised manuscript.

Fig 1. Provincial spatial distribution of subjective poverty in 2019

Fig 2. Provincial spatial distribution of public service supply in 2019

Fig 3. Provincial spatial distribution of physical capital in 2019

Fig 4. Provincial spatial distribution of financial capital in 2019

Fig 5. Provincial spatial distribution of human capital in 2019

Fig 6 Provincial spatial distribution of social capital in 2019

Reviewer #2: 

The paper is recommended for publication as it is a valuable contribution, and relevant with respect to the stated goal which is fully achieved. I recommend the publication of this paper in its current form.

Response:

We are very grateful to the reviewers for recognizing and supporting our manuscript. It makes us feel very honored and happy, and we will continue to work hard and deepen our research in the future.

Thank you again for taking the time to review our manuscript and provide valuable feedback. We greatly appreciate your expertise and insightful comments, which have undoubtedly improved the quality of our research. We also extend our gratitude to the editors for their assistance throughout the review process. Thank you for your support and guidance. We look forward to hearing from you regarding our submission. We would be glad to respond to any further questions and comments that you may have.

Best wishes to you.

Kind regards,

Yuanquan Lu, Li Chen, Yuan Meng

---

## [Decision Letter · Decision Letter 2]

27 Sep 2023

How do public services supply, livelihood capital, and livelihood strategies affect subjective poverty?

PONE-D-23-13234R2

Dear Dr. Chen,

We’re pleased to inform you that your manuscript has been judged scientifically suitable for publication and will be formally accepted for publication once it meets all outstanding technical requirements.

Kind regards,

Chaohai Shen

Academic Editor

PLOS ONE

Additional Editor Comments (optional):

Reviewers' comments:

Reviewer's Responses to Questions

**Comments to the Author**

1. If the authors have adequately addressed your comments raised in a previous round of review and you feel that this manuscript is now acceptable for publication, you may indicate that here to bypass the “Comments to the Author” section, enter your conflict of interest statement in the “Confidential to Editor” section, and submit your "Accept" recommendation.

Reviewer #1: All comments have been addressed

2. Is the manuscript technically sound, and do the data support the conclusions?

Reviewer #1: Yes

3. Has the statistical analysis been performed appropriately and rigorously? 

Reviewer #1: Yes

4. Have the authors made all data underlying the findings in their manuscript fully available?

Reviewer #1: Yes

5. Is the manuscript presented in an intelligible fashion and written in standard English?

Reviewer #1: Yes

6. Review Comments to the Author

Reviewer #1: According to the criteria of the reviewers' remarks, the authors did a decent job of revising the manuscript. I advise you to accept the document after thoroughly checking it for typos.

7. PLOS authors have the option to publish the peer review history of their article (what does this mean?). If published, this will include your full peer review and any attached files.

Reviewer #1: No

---

## [Editor Report · Acceptance letter]

28 Sep 2023

PONE-D-23-13234R2 

How do public services supply, livelihood capital, and livelihood strategies affect subjective poverty? 

Dear Dr. Chen:

I'm pleased to inform you that your manuscript has been deemed suitable for publication in PLOS ONE. Congratulations! Your manuscript is now with our production department. 

Kind regards, 

on behalf of

Dr. Chaohai Shen 

Academic Editor

PLOS ONE